# COVID-19 and Tuberculosis: Unveiling the Dual Threat and Shared Solutions Perspective

**DOI:** 10.3390/jcm12144784

**Published:** 2023-07-19

**Authors:** Ramona Cioboata, Viorel Biciusca, Mihai Olteanu, Corina Maria Vasile

**Affiliations:** 1Department of Pneumology, University of Pharmacy and Medicine Craiova, 200349 Craiova, Romania; ramona_cioboata@yahoo.com; 2Department of Pneumology, Victor Babes Clinical Hospital, 030303 Craiova, Romania; 3Department of Internal Medicine, Filantropia Hospital, 050474 Craiova, Romania; 4Department of Pediatric and Adult Congenital Cardiology, Bordeaux University Hospital, 33600 Pessac, France; corina.vasile93@gmail.com

**Keywords:** COVID-19, tuberculosis, dual-threat, quality of life, pandemic, perspective

## Abstract

The year 2020 will likely be remembered as the year dominated by COVID-19, or coronavirus disease. The emergence of severe acute respiratory syndrome coronavirus 2 (SARS-CoV-2), responsible for this pandemic, can be traced back to late 2019 in China. The COVID-19 pandemic has significantly impacted the tuberculosis (TB) care system, reducing TB testing and reporting. This can be attributed to the disruption of TB services and restrictions on patient movement, consequently increasing TB-related deaths. This perspective review aims to highlight the intersection between COVID-19 and TB, highlighting their dual threat and identifying shared solutions to address these two infectious diseases effectively. There are several shared commonalities between COVID-19 and tuberculosis, particularly the transmission of their causative agents, severe acute respiratory syndrome coronavirus 2 (SARS-CoV-2) and Mycobacterium tuberculosis. Both pathogens are transmitted via respiratory tract secretions. TB and COVID-19 are diseases that can be transmitted through droplets and airborne particles, and their primary target is typically the lungs. Regarding COVID-19 diagnostics, several methods are available for rapid and accurate detection. These include RT-PCR, which can provide results within two hours, and rapid antigen test kits that offer results in just a few minutes. The availability of point-of-care self-testing further enhances convenience. On the other hand, various approaches are employed for TB diagnostics to swiftly identify active TB. These include sputum microscopy, sputum for reverse transcription polymerase chain reaction (RT-PCR), and chest X-rays. These methods enable the rapid detection of active TB on the same day, while culture-based testing may take significantly longer, ranging from 2 to 8 weeks. The utilization of diverse diagnostic tools helps ensure the timely identification and management of COVID-19 and TB cases. The quality of life of patients affected by COVID-19 and tuberculosis (TB) can be significantly impacted due to the nature of these diseases and their associated challenges. In conclusion, it is crucial to emphasize the urgent need to address the dual threat of COVID-19 and TB. Both diseases have devastated global health, and their convergence poses an even greater challenge. Collaborative efforts, research investments, and policy reforms are essential to tackle this dual threat effectively.

## 1. Introduction

The global healthcare systems have faced unprecedented challenges recently due to the COVID-19 pandemic, causing widespread disruptions in societies worldwide. While the focus has understandably been on combating COVID-19, it is crucial not to overlook the long-standing burden of tuberculosis.This highly infectious disease has afflicted humanity for centuries.

The year 2020 will probably be remembered as the year dominated by COVID-19, or coronavirus disease. The emergence of severe acute respiratory syndrome coronavirus 2 (SARS-CoV-2), responsible for this pandemic, can be traced back to late 2019 in China [1,2,3]. While COVID-19 remains prominent in scientific literature and media coverage, it is important not to neglect other communicable diseases, including tuberculosis [4].

The COVID-19 pandemic has significantly impacted the TB care system, reducing TB testing and reporting. This can be attributed to the disruption of TB services and restrictions on patient movement, consequently increasing TB-related deaths [5,6,7,8,9]. This perspective review aims to highlight the intersection between COVID-19 and TB, highlighting their dual threat and identifying shared solutions to address these two infectious diseases effectively.

## 2. Epidemiological Overlap

The COVID-19 pandemic and TB share several striking similarities in terms of epidemiological characteristics. Both diseases are primarily transmitted through respiratory droplets, making close contact and crowded environments conducive to their spread. Additionally, marginalized and vulnerable populations, including those with weakened immune systems, the elderly, and individuals with underlying health conditions, are particularly susceptible to COVID-19 and TB.

The impact of the COVID-19 pandemic on TB is multifaceted. Health resources have been diverted to combat the new threat, disrupting TB diagnosis, treatment, and control efforts. Lockdowns, travel restrictions, and reduced access to healthcare facilities have impeded TB case finding and hindered timely diagnosis. The convergence of these two infectious diseases has created a complex situation demanding immediate attention and comprehensive strategies.

There are several shared commonalities between COVID-19 and tuberculosis, particularly the transmission of their causative agents, severe acute respiratory syndrome coronavirus 2 (SARS-CoV-2) and *Mycobacterium tuberculosis* [10]. Both pathogens are transmitted via respiratory tract secretions [11,12,13,14]. TB and COVID-19 are diseases that can be transmitted through droplets and airborne particles, and their primary target is typically the lungs. However, it is important to note that these diseases can potentially affect various organs in the body [15,16,17]. Furthermore, protecting healthcare workers and other susceptible patients and identifying and assessing contact are key components of the public health response to both infections. Understanding the pathways and factors influencing transmission is necessary to develop effective and efficient disease control measures. For tuberculosis, many years of clinical and experimental studies have provided a wealth of information on which to base the contact’s identification, prioritization, and assessment [11]. Not surprisingly, this knowledge of SARS-CoV-2 transmission needs to be improved, and the relative contributions to the transmission of large respiratory droplets, fomites, and aerosols remain controversial [16]. Notably, the transmission of both pathogens has been associated with over-extension events [17,18,19,20]. A diagram presenting the most frequent clinical signs and the multi-organ involvement is presented below in Figure 1.

Both tuberculosis (TB) and COVID-19 have significant pulmonary involvement, making the respiratory system a major battleground for these diseases. Tuberculosis primarily affects the lungs, leading to progressive lesions, chronic cough, and respiratory symptoms. It can cause extensive destruction of lung tissue and form caverns, leading to persistent coughing and a potential spread of infectious droplets [21,22]. Similarly, COVID-19 primarily targets the respiratory system, causing severe pneumonia and acute respiratory distress syndrome (ARDS). This virus damages lung cells, triggering inflammation and compromising respiratory function [23,24,25]. The overlapping respiratory manifestations of tuberculosis and COVID-19 highlight the critical importance of understanding and managing the respiratory dynamics of these diseases to prevent transmission, ensure timely diagnosis, and develop effective treatment strategies [26].

Both COVID-19 and tuberculosis share an imbalance of immune responses based on individual immunological mechanisms. This suggests that co-infection may increase the risk of disease progression in both diseases [27]. Chronic lung disease, diabetes mellitus, smoking, and liver failure are just a few of the medical factors that increase the risk of severe illness and the need for intensive care units or mechanical ventilation associated with COVID-19. Once tuberculosis has been present in a patient’s body, tobacco smoking increases the chances of inadequate therapy, leading to delayed sputum culture and treatment. There is growing evidence of an increased probability of COVID-19 in people with diabetes, which can lead to hospitalization, organ failure, and premature mortality [28]. In patients with tuberculosis and COVID-19, malnutrition and low body mass index (BMI) are major risk factors for early mortality [29].

COVID-19 has a shorter incubation period (1–14 days), while the TB incubation period lasts two weeks to several years before active TB develops.

COVID-19 is characterized by symptoms such as cough, fever, difficulty breathing, sore throat, reduced or loss of smell, taste loss, diarrhea, muscle pain, and fatigue. These symptoms typically appear abruptly. In contrast, TB presents with symptoms including fever, night sweats, ongoing productive cough, coughing up blood, loss of appetite, chest pain, and fatigue. However, the onset of TB symptoms is insidious, with gradual development.

In the case of COVID-19, severe presentations are often observed in individuals with comorbidities such as human immunodeficiency virus (HIV), chronic lung disease, chronic heart conditions, obesity, an immunocompromised state, and diabetes mellitus. These underlying health conditions can contribute to a more severe disease course. On the other hand, for TB, comorbidities such as diabetes mellitus, sickle cell disease, chronic lung disease, HIV, and an immunocompromised state are known to increase the risk and severity of TB infection. Recognizing and managing these comorbidities are crucial in preventing and addressing severe presentations of both COVID-19 and TB [30,31]. The most common risk factors for COVID-19 and tuberculosis are presented in Figure 2.

Regarding COVID-19 diagnostics, several methods are available for rapid and accurate detection. These include RT-PCR, which can provide results within two hours, and rapid antigen test kits that offer results in just a few minutes. The availability of point-of-care self-testing further enhances convenience [32,33]. On the other hand, various approaches are employed for TB diagnostics to swiftly identify active TB. These include sputum microscopy, sputum RT-PCR, and chest X-rays. These methods enable the rapid detection of active TB on the same day, while culture-based testing may take significantly longer, ranging from 2 to 8 weeks. The utilization of diverse diagnostic tools helps ensure the timely identification and management of COVID-19 and TB cases [34].

COVID-19 samples are typically collected through naso- and oropharyngeal swabs and saliva, which are relatively easy-to-obtain specimens. In contrast, for tuberculosis testing, sputum or extrapulmonary samples are required for accurate diagnosis. Collecting appropriate samples is essential to effectively detect and diagnose COVID-19 and TB cases [35].

COVID-19 sequelae can result in various long-term effects, including cognitive impairments, mental health disorders, respiratory disorders, ageusia (loss of taste), and hypo/anosmia (reduced or loss of smell). In the case of TB sequelae, residual lung disease characterized by bronchiectasis, scars, and cavities is frequently observed. These sequelae can lead to reduced pulmonary function and recurrent infections. Recognizing and addressing these long-term effects are crucial in providing comprehensive care to individuals affected by COVID-19 and TB [36,37,38,39].

We have highlighted the most important data in Figure 3 for a concise summary of this subsection.

## 3. Impact on Diagnosis and Screening

The COVID-19 pandemic has profoundly impacted TB diagnosis and screening programs, posing significant challenges to the global effort to combat tuberculosis. The diversion of resources, reduced access to healthcare facilities, and disruptions in diagnostic services have hampered TB case finding and early detection during these unprecedented times.

The overwhelming focus on combating COVID-19 has led to the diversion of financial and human resources from TB programs. Once allocated to TB diagnosis and screening, funding has been redirected towards COVID-19 response efforts, leaving TB programs under-resourced. This diversion has reduced the capacity to conduct TB screenings and delayed the implementation of TB control strategies.

Lockdown measures, travel restrictions, and overwhelmed healthcare systems have created barriers to accessing healthcare facilities for individuals seeking TB diagnosis and screening. Patients with TB symptoms may hesitate to visit healthcare centers due to fears of contracting COVID-19 or concerns about overwhelming healthcare facilities already burdened by the pandemic. The reduced accessibility has resulted in missed opportunities for early TB case detection [40].

COVID-19 has caused disruptions in laboratory services, including shortages of diagnostic supplies, reduced staffing levels, and increased turnaround times for test results. These disruptions have significantly affected the availability and timeliness of TB diagnostic services. Patients may experience delays in receiving their test results, leading to delayed initiation of treatment and increased risk of TB transmission within communities [41].

Increasing community awareness about TB symptoms, transmission, and early diagnosis is crucial. Intensified community outreach and education campaigns, leveraging various media channels, can empower individuals to recognize TB symptoms and seek healthcare promptly, even during the pandemic [42,43].

Efforts should be made to integrate TB and COVID-19 diagnostic services wherever feasible. This integration could include joint screening programs, shared laboratory resources, and co-location of services. By leveraging existing COVID-19 infrastructure, such as testing centers and sample collection sites, TB case finding can be enhanced while optimizing resource utilization [44].

Telemedicine platforms can be vital in remote screening, diagnosis, and monitoring of individuals with suspected TB. Virtual consultations and digital media can enable healthcare providers to assess symptoms, provide guidance, and order necessary diagnostic tests. Telemedicine can help overcome barriers to access while reducing the risk of COVID-19 exposure [45]. Telemedicine is a valuable technology that connects doctors and patients to ensure long-term lifestyle changes. It has significant benefits for staff in doctors’ surgeries. It often removes the burden of patient registration and focuses on higher-value tasks. With the ability to visit online, doctors can take care of their patients while potentially helping other affected practices simultaneously. This also reduces distance-related limitations by electronically sharing information about a diagnosis, care, and disease prevention between doctor and patient. Expanded telemedicine can bring medical coverage closer to people living in rural areas where quality treatment is impossible. In recent years, this technology has been shown to increase the quality of healthcare services by allowing information to be exchanged in very remote areas. It extends access to underserved areas, making it easier for people in these areas to make appointments and consultations. People with reduced mobility receive the medical advice and prescriptions they need more quickly—medications and tests and procedures they need to manage in place. Telemedicine minimizes doctors’ and patients’ travel around the globe and makes a difference in every patient’s life, ensuring that every patient receives the right medical treatment [46].

Exploring innovative diagnostic approaches, such as molecular tests and point-of-care diagnostics, can enhance early TB detection. These technologies offer rapid results, reducing testing and treatment initiation time. Investing in the research and development of cost-effective and user-friendly diagnostic tools can strengthen TB case finding during the pandemic and beyond [47,48].

Effective data collection and surveillance systems are vital for monitoring TB cases and identifying areas with high transmission rates. Surveillance data can guide targeted interventions and resource allocation. Leveraging digital health platforms and data analytics can provide real-time insights and facilitate proactive decision-making [49].

The COVID-19 pandemic has profoundly affected the epidemiology of tuberculosis (TB), particularly through its impact on unemployment, poverty, and malnutrition. The widespread economic disruption caused by the pandemic has significantly increased global unemployment rates [50,51].

Limited economic resources and income have pushed many individuals and families into poverty, exacerbating already difficult socio-economic conditions, which are known risk factors for TB transmission and disease progression [52]. As people struggle to meet their basic needs, malnutrition becomes more widespread, weakening the immune system and increasing susceptibility to TB infection [53]. 

Studies consistently show a strong link between poverty, malnutrition, and the increased incidence of TB. The indirect consequences of the COVID-19 pandemic, such as increased poverty and malnutrition, have created fertile ground for TB to persist and flourish [29,54]. To effectively combat TB in the post-pandemic era, addressing health’s social and economic determinants is essential, focusing on reducing poverty, improving nutrition, and ensuring access to health services for vulnerable populations.

## 4. Co-Infection and Clinical Management

Co-infection with COVID-19 and TB presents unique challenges in clinical management. The interaction between these two infectious diseases can have significant implications, including potential drug interactions, immune dysregulation, and increased morbidity and mortality rates [55]. This section explores the clinical implications of co-infection and emphasizes the importance of integrated care and collaborative approaches to optimize patient outcomes.

Patients with co-infection require careful consideration of the drug regimens for both COVID-19 and TB. Certain medications used to treat COVID-19, such as protease inhibitors or remdesivir, can interact with drugs commonly prescribed for TB, such as rifampicin or isoniazid. Drug interactions can affect the efficacy and safety of treatment for both diseases, necessitating close monitoring and possible adjustments to medication regimens. Clinicians must strike a balance between managing co-infection and avoiding adverse drug reactions.

All patients diagnosed with active TB are treated with a multidrug regimen known as antitubercular therapy. For patients with TB caused by pan-susceptible organisms, the treatment regimen consists of isoniazid, rifampin, pyrazinamide, and ethambutol [56,57].

Various therapeutic agents have been employed to treat COVID-19 in patients with TB co-infection. These include hydroxychloroquine, azithromycin, lopinavir/ritonavir, and the darunavir/cobicistat combination [57,58]. Additionally, glucocorticoids such as methylprednisolone and dexamethasone are administered to specific subsets of patients. Some reports also indicated the use of anticoagulation medications like enoxaparin and parnaparine [58]. 

It is important to note that while the antitubercular regimens used for TB treatment are similar among patients, the therapies used for COVID-19 vary [59,60].

COVID-19 and TB can cause immune dysregulation, further complicating clinical management. The immune response in individuals coinfected with COVID-19 and tuberculosis (TB) is intricate and influenced by innate and adaptive immunity. Current evidence suggests that in patients with latent TB infection (LTBI), there is a positive immunomodulation against COVID-19, likely because of trained innate immunity and crossed heterologous immunity. In contrast, patients with active TB (aTB) may experience a reduced specific response to SARS-CoV-2 and decreased lymphocyte function, which could lead to ineffective control of the infection.

Co-infection may exacerbate immune dysregulation, leading to more severe disease manifestations and poorer clinical outcomes. Monitoring and managing the immune response in individuals with co-infection are crucial to minimize complications and optimize treatment outcomes [61].

Diagnosing co-infection can be challenging due to overlapping symptoms and diagnostic difficulties. COVID-19 can present with respiratory symptoms similar to those of TB, such as cough and shortness of breath. The diagnostic tests for COVID-19 and TB differ, with molecular tests for SARS-CoV-2 and sputum or molecular tests for TB. Clinicians must consider the possibility of co-infection when evaluating individuals with respiratory symptoms, ensuring timely and accurate diagnoses for both diseases.

A collaborative and integrated approach to care is essential to optimize patient outcomes. Coordinating efforts between healthcare providers specializing in infectious diseases, pulmonology, and critical care is crucial for managing co-infected individuals effectively. Integrated care pathways and multidisciplinary teams can ensure seamless communication and coordinated treatment plans, considering the unique challenges posed by co-infection. Shared decision-making and the close monitoring of treatment responses and potential adverse events are vital for comprehensive management.

Long-term follow-up care is crucial for individuals recovering from co-infection. COVID-19 and TB can have long-lasting effects on respiratory function and overall health. The regular monitoring of lung function, radiological imaging, and microbiological assessments are necessary to detect and manage potential relapses or complications. Comprehensive post-infection care should address physical, psychological, and social aspects to support individuals in their recovery journey.

Co-infection with COVID-19 and TB presents complex challenges in clinical management. The potential for drug interactions, immune dysregulation, diagnostic difficulties, and long-term complications necessitates a holistic and collaborative approach to optimize patient outcomes. Integrated care, close monitoring, and multidisciplinary collaboration among healthcare providers are crucial in managing co-infected individuals effectively. By addressing these challenges collectively, healthcare systems can enhance patient care, mitigate the disease burden, and improve the long-term prognosis of individuals with co-infection.

## 5. Lessons Learned

The COVID-19 pandemic has highlighted the critical importance of a robust and coordinated response to infectious diseases. While COVID-19 and tuberculosis (TB) differ in many aspects, there are areas of synergy between the responses to these diseases. This section explores the potential synergies and emphasizes the importance of leveraging lessons learned from the COVID-19 pandemic to strengthen TB control efforts. It discusses integrating surveillance systems, research collaborations, and public health strategies to address both diseases simultaneously.

Integrating surveillance systems for COVID-19 and TB can yield valuable synergies. The infrastructure and data collection mechanisms established for COVID-19 can be leveraged to strengthen TB surveillance, enabling real-time monitoring of TB cases and facilitating early detection and response. By integrating data systems, healthcare providers and public health authorities can identify overlapping populations, assess the impact of COVID-19 on TB control efforts, and tailor interventions accordingly.

Research collaborations established during the COVID-19 pandemic can be expanded to include TB research. Lessons learned from COVID-19 research, such as clinical trials, epidemiological studies, and genomic sequencing, can inform TB research and vice versa. Collaborative efforts can expedite the development of new diagnostics, treatments, and vaccines for both diseases. Furthermore, shared research platforms and networks can enhance knowledge exchange, capacity building, and innovation in infectious diseases.

The COVID-19 pandemic has exposed weaknesses in health systems globally. Addressing these weaknesses presents an opportunity to strengthen the capacity of health systems for TB control. Investments in healthcare infrastructure, laboratory networks, supply chains, and healthcare workforce training can benefit both COVID-19 and TB responses. Strengthening health systems can enhance diagnostic capabilities, improve access to care, and support the integration of services to manage both diseases.

Lessons learned from COVID-19 can inform public health strategies for TB control. Strategies such as infection prevention and control measures, community engagement, risk communication, and contact tracing have proven effective in controlling the spread of COVID-19. They can be adapted for TB control efforts. Integrating public health programs and resources can optimize resource allocation, improve efficiency, and enhance the overall effectiveness of disease control initiatives.

The COVID-19 pandemic has highlighted existing health inequities, including disparities in access to healthcare, socioeconomic factors, and vulnerable populations disproportionately affected by infectious diseases. Addressing health inequities is crucial for both COVID-19 and TB control efforts. By prioritizing equity in resource allocation, testing strategies, treatment access, and public health interventions, synergistic efforts can lead to more equitable outcomes in combating both diseases.

The response to the COVID-19 pandemic has provided valuable lessons that can strengthen TB control efforts. Leveraging synergies between the two diseases, such as integrated surveillance systems, research collaborations, strengthening health systems, adopting effective public health strategies, and addressing health inequities, can optimize the response to both COVID-19 and TB. By integrating efforts, sharing knowledge, and capitalizing on the momentum created during the pandemic, we can accelerate progress toward ending the TB epidemic while effectively addressing emerging infectious diseases in the future.

An additional important lesson learned from the COVID-19 pandemic provided an opportunity to integrate innovative approaches, such as telemedicine, into TB services. The pandemic required a rapid shift to virtual healthcare delivery to ensure the continuity of care while minimizing in-person interactions. Telemedicine, including teleconsultations and telemonitoring, has proven effective in facilitating remote patient management, ensuring access to healthcare services, and improving adherence to treatment protocols. These advances in technology and digital health solutions can be exploited to strengthen TB services in the future. Telemedicine can enable the remote diagnosis and follow-up of TB patients, reducing the need for in-person visits and overcoming geographical barriers. It also provides a platform for healthcare providers to educate and support patients, empowering them to participate in their care actively [62,63,64,65].

In addition, telemedicine can improve collaboration and knowledge sharing between healthcare professionals, allowing multidisciplinary teams to discuss complex cases and exchange expertise across different regions. By integrating telemedicine into TB services, we can optimize resource allocation, improve patient outcomes, and increase the overall effectiveness of TB control programs. As we navigate the post-pandemic era, we must embrace these lessons from COVID-19 and leverage telemedicine as a valuable tool to transform and improve the delivery of TB care [66,67,68].

## 6. Looking Forward: Post-Pandemic Challenges for Tuberculosis and Advancements in Vaccine Development

While the COVID-19 pandemic begins to unravel, we must focus on addressing the continuing impact of tuberculosis (TB). As the world struggles with the unique challenges posed by COVID-19, it is critical that we maintain our focus on the long-term burden of TB, which has persisted for years [69,70]. 

TB remains a significant global health problem, with millions of new cases and deaths reported annually. The pandemic has disrupted TB control efforts, leading to delays in diagnosis, treatment interruptions, and the exacerbation of existing challenges. In addition, the indirect consequences of the COVID-19 pandemic, such as weakened health systems and disruption of healthcare services, may further exacerbate the burden of TB [71]. 

It is, therefore, essential to recognize the enduring importance of combating TB and to ensure that efforts to control this infectious disease are sustained and strengthened even as the focus shifts to the post-pandemic era.

The development of vaccines has played a key role in public health, and the recent attention and remarkable progress in the development of COVID-19 vaccines highlight the potential for scientific progress. However, it is important to recognize that TB, a disease that has affected humanity for many years, still lacks effective preventive measures [72,73]. The Bacillus Calmette-Guérin (BCG) vaccine, developed almost a century ago, remains the main vaccine for TB prevention [74,75]. However, its effectiveness varies greatly depending on the population and does not provide complete protection against the disease. Despite the enormous need for more effective TB vaccines, this area of research has received considerably less attention and investment than COVID-19 [76,77]. 

The ongoing global pandemic has undoubtedly accelerated the research efforts and resources devoted to COVID-19 vaccines, highlighting the importance of prioritizing the research and development of vaccines for other infectious diseases, such as TB, to address the persistent global burden they pose.

## 7. Life Quality

### 7.1. During Infection

The quality of life of patients affected by COVID-19 and TB can be significantly impacted due to the nature of these diseases and their associated challenges.

For COVID-19 patients, the severity of the illness can vary widely. Those with mild or moderate symptoms may experience fatigue, cough, shortness of breath, and loss of taste or smell. Although these symptoms can cause discomfort and hinder daily activities, they often resolve within a few weeks. However, the impact on quality of life can be much more pronounced for individuals with severe or critical cases. Hospitalization, sometimes in intensive care units, and the potential need for mechanical ventilation can lead to physical and emotional distress. Prolonged recovery periods may be required, and some patients may experience long-term effects on their respiratory function, exercise tolerance, and overall physical well-being. In addition, cognitive impairments, such as brain fog and memory issues, have been reported in some individuals, further affecting their quality of life. Mental health issues, including anxiety, depression, and post-traumatic stress disorder, can also arise due to the traumatic experience of severe illness and hospitalization [78].

In the case of TB patients, the disease can have a significant impact on quality of life. Persistent coughing that is often accompanied by sputum production, chest pain, fatigue, and weight loss are common symptoms that can cause physical discomfort and interfere with daily activities. TB treatment, which typically involves a prolonged course of antibiotics for several months, can have side effects, such as gastrointestinal problems and liver toxicity, and may require strict adherence to medication regimens. These treatment requirements can be burdensome for patients, impacting their ability to work, engage in social activities, and lead fulfilling lives. Additionally, the stigma associated with TB, fueled by misconceptions about the disease’s transmission, can lead to social isolation and discrimination, further affecting the quality of life for individuals with the disease [79].

It is important to note that COVID-19 and TB can have long-term consequences, and the impact on quality of life may extend beyond the acute phase of the illness. Rehabilitation and support services, including physical therapy, respiratory therapy, and psychological counseling, are crucial to addressing patients’ challenges and improving their overall well-being. Mental health interventions and support systems are also essential to help individuals cope with the psychological impact of these diseases. Moreover, community education and awareness programs are vital in combating the stigma associated with TB, fostering understanding and promoting the acceptance of individuals who have experienced the disease [80].

By recognizing the multifaceted impact of these diseases on quality of life and implementing comprehensive care approaches, healthcare systems can strive to improve the well-being of COVID-19 and TB patients, supporting their recovery and helping them regain a satisfactory quality of life.

### 7.2. After Infection

Post-COVID-19 and TB, patients may experience various challenges impacting their quality of life. While some individuals may recover fully and resume normal activities, others may face lingering effects and long-term complications [81].

For post-COVID-19 patients, a condition known as long COVID, or post-acute sequelae of SARS-CoV-2 infection (PASC), can arise. This condition is characterized by persistent symptoms that last beyond the acute phase of the illness. Fatigue, shortness of breath, joint and muscle pain, cognitive difficulties, and mental health issues such as anxiety and depression are common manifestations of long COVID. These symptoms can significantly impact a person’s ability to carry out daily activities, return to work, and engage in social interactions. Long COVID may require ongoing medical care, rehabilitation, and support services to help patients manage their symptoms and regain their quality of life [82,83].

Similarly, TB survivors may face challenges after completing their treatment. Residual lung damage, such as bronchiectasis, scarring, and cavities, can reduce pulmonary function and increase susceptibility to respiratory infections. Persistent respiratory symptoms, including coughing and shortness of breath, may persist even after successful treatment. These respiratory complications can limit physical activities and impact overall well-being. TB survivors may also experience social and psychological difficulties due to the stigma associated with the disease. Overcoming these challenges and reintegrating into society may require ongoing support, counseling, and efforts to combat the TB-related stigma [72,84,85].

Both post-COVID-19 and post-TB individuals may benefit from multidisciplinary care approaches that address their specific needs. Rehabilitation programs, including physical therapy and respiratory exercises, can help improve lung function, restore strength, and enhance overall physical well-being. Psychological support, such as counseling and mental health interventions, may be necessary to address anxiety, depression, and other mental health issues that can arise from the experience of these diseases. Additionally, patient education, support groups, and community engagement can promote understanding, reduce stigma, and provide a supportive environment for individuals recovering from COVID-19 and TB [86,87].

Healthcare systems need to recognize and address the long-term consequences of these diseases to ensure comprehensive post-treatment care and support for affected individuals. Providing appropriate medical interventions, rehabilitation, and psychosocial support can enhance the quality of life and facilitate the successful reintegration of post-COVID-19 and post-TB patients into society [88].

Figure 4 depicts the most common aspects that can impact the quality of life in patients who have experienced COVID-19 or tuberculosis infection.

## 8. Conclusions

In conclusion, it is crucial to emphasize the urgent need to address the dual threat of COVID-19 and TB. Both diseases have devastated global health, and their convergence poses an even greater challenge. Collaborative efforts, research investments, and policy reforms are essential to tackle this dual threat effectively.

First and foremost, collaborative efforts among governments, international organizations, healthcare providers, and communities are necessary to develop comprehensive strategies. The fight against COVID-19 and TB requires a coordinated approach encompassing prevention, diagnosis, treatment, and support for those affected. By working together, we can pool resources, share expertise, and implement best practices, thereby increasing the effectiveness of our responses.

Furthermore, research investments are vital to understand the interactions between COVID-19 and TB and to develop innovative solutions. Both diseases are complex, and their interplay may have far-reaching implications. Robust research can provide insights into the impact of COVID-19 on TB cases, the effectiveness of existing TB treatments in COVID-19 patients, and the development of vaccines that offer dual protection. By investing in research, we can arm ourselves with the knowledge needed to combat these diseases more effectively.

Policy reforms are also necessary to strengthen health systems and ensure comprehensive responses to COVID-19 and TB. This includes allocating adequate resources, improving healthcare infrastructure, enhancing diagnostic capabilities, and promoting access to quality care. Additionally, policy reforms should prioritize vulnerable populations disproportionately affected by both diseases and ensure that healthcare services are equitable and accessible to all.

Importantly, addressing the dual threat of COVID-19 and TB presents an opportunity to find shared solutions and build a more resilient and equitable global health landscape. The lessons learned from the COVID-19 pandemic can be applied to strengthen TB control measures and vice versa. Investments in health systems, infrastructure, and human resources will benefit the response to these diseases and future health crises. We can create a more just and equitable world by taking a holistic approach and addressing the root causes of health disparities.

In conclusion, the urgent need to address the dual threat of COVID-19 and TB cannot be overstated. We can strengthen health systems and mount comprehensive responses to both diseases through collaborative efforts, research investments, and policy reforms. This is an opportunity to build a more resilient and equitable global health landscape where no one is left behind in the fight against infectious diseases. Let us seize this moment and work together towards a healthier future.

## Figures and Tables

**Figure 1 jcm-12-04784-f001:**
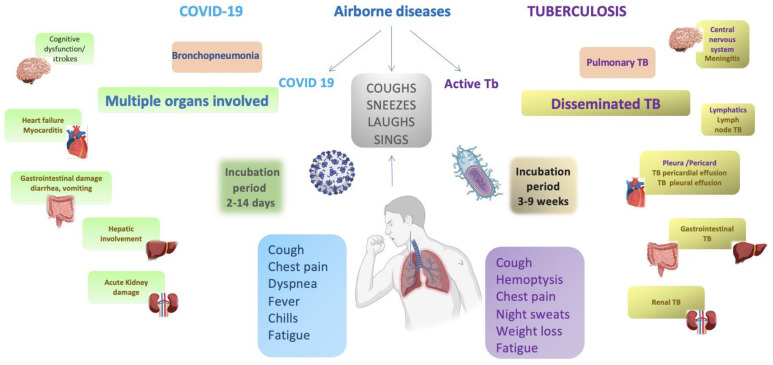
Common symptoms and multi-organ involvement of COVID-19 and TB.

**Figure 2 jcm-12-04784-f002:**
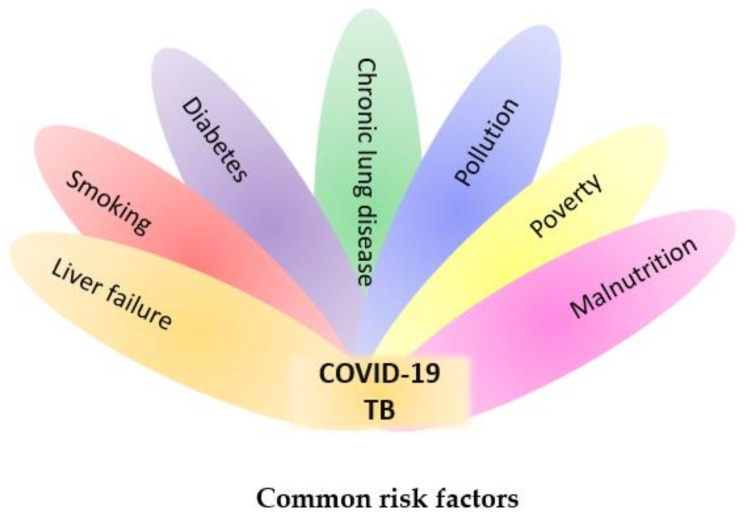
Common risk factors for COVID-19 and tuberculosis.

**Figure 3 jcm-12-04784-f003:**
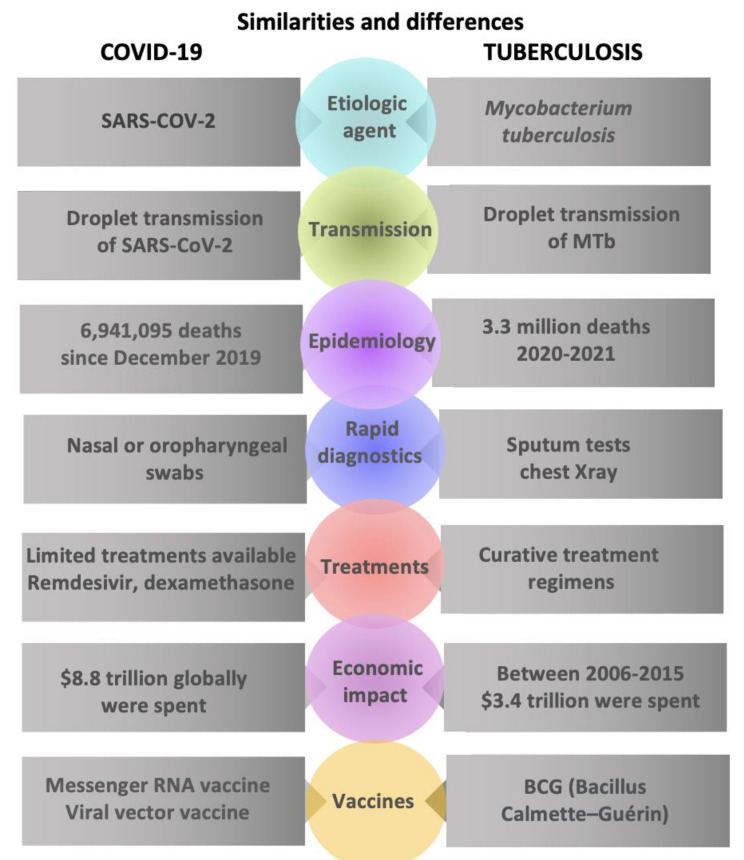
Overall similarities and differences between COVID-19 and tuberculosis.

**Figure 4 jcm-12-04784-f004:**
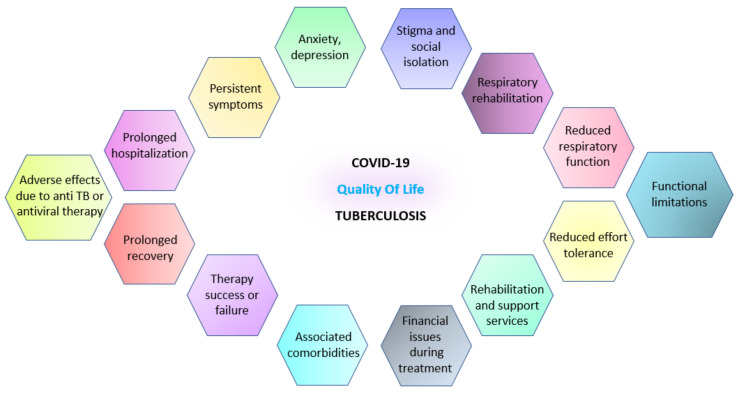
Life quality-shared manifestations for COVID-19 and tuberculosis while experiencing the infection and after.

## Data Availability

More data are available on request from the corresponding authors.

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
