# Peer review of "COVID-19 and Tuberculosis: Unveiling the Dual Threat and Shared Solutions Perspective"

_jcm, 2023, doi:10.3390/jcm12144784_

Round 1
Reviewer 1 Report
Dear Authors
This is an interesting article because it is an update on TB and Covid-19. You present a comprehensive overview of the two diseases with recent data from the literature. This is a well-written text that is of interest to health professionals and students due to the topic's opportunity.
I only suggest emphasizing the pulmonary involvement of the two diseases in the text that precedes Figure 1 and in the figure itself (although the lungs have been highlighted in the color of the drawing). The extra pulmonary manifestations ended up being more prominent than the pulmonary ones.
Author Response
Dear Reviewer,
We highly appreciate your kind words.
Following your recommendations, we modified it by adding a paragraph after Figure 1.
Reviewer 2 Report
This is an interesting well written review on the interaction of the two most lethal infectious diseases of our time. Many different aspects are discussed. My main comments are:
Figure 3. Vaccines for both diseases can be added and the attention given and speed of development along with the efficiency of covid vaccines can be compared to TB in the text – BCG is 100 year old vaccine, and the need for new more effective vaccines for TB is enormous. However TB vaccines have received much less attention than Covid.
Figure 3. what does the economic impact refer to? The money spent or needed? TB has received much less attention and budget than needed over the years in contrast to Covid. A possible explanation may be that TB affects many countries under development whereas Covid has mainly affected the developed world. The authors can analyze this discrepancy
The authors may want to emphasize that lessons learnt from Covid such as telemedicine can be intergraded in the future in TB services
The authors may want to describe the effect of Covid on TB epidemiology by increasing unemployment and therefore poverty and malnutrition which are well documented risk factors for TB
As the Covid pandemic is fading away the authors may also discuss the continuous impact of TB.
Minor comments
Figure 3. the comparison of deaths could improve if the number of deaths for TB for years 2020, 2021 and 2022 is included rather than 2021 alone
Figure 1. The significant difference in the incubation period may be included in the figure
Lines 40-41: While 40 the focus has understandably been on combating COVID-19. It is crucial not to overlook the long-standing burden of tuberculosis (TB). The two sentences should merged into one
Line 70 and Figure 3: please use italics for Mycobacterium tuberculosis
Figure 1. disseminated TB, pericardium- instead of disseminated and pericard
Author Response
Dear reviewer,
We highly appreciate your kind words.
development along with the efficiency of covid vaccines can be compared to TB in the text – BCG is 100 year old vaccine, and the need for new more effective vaccines for TB is enormous. However TB vaccines have received much less attention than Covid.
We have improved Figure 3. In addition, we have added a paragraph for Covid vaccines.
Figure 3. what does the economic impact refer to? The money spent or needed? TB has received much less attention and budget than needed over the years in contrast to Covid. A possible explanation may be that TB affects many countries under development whereas Covid has mainly affected the developed world. The authors can analyze this discrepancy
The authors may want to emphasize that lessons learnt from Covid such as telemedicine can be intergraded in the future in TB services
We have addressed this.
The authors may want to describe the effect of Covid on TB epidemiology by increasing unemployment and therefore poverty and malnutrition which are well documented risk factors for TB
We have discussed the effect of COVID on TB epidemiology as you have suggested.
As the Covid pandemic is fading away the authors may also discuss the continuous impact of TB.
We have addressed this in a short paragraph.
Minor comments
Figure 3. the comparison of deaths could improve if the number of deaths for TB for years 2020, 2021 and 2022 is included rather than 2021 alone.
Regarding mortality data, now there are data available from 2020 and 2021. The WHO 2022 report will appear in March 2024 (the reports are delayed by 2 years).
Figure 1. The significant difference in the incubation period may be included in the figure.
We have added the missing data.
Lines 40-41: While the focus has understandably been on combating COVID-19. It is crucial not to overlook the long-standing burden of tuberculosis (TB). The two sentences should merged into one
We have revised it.
Line 70 and Figure 3: please use italics for Mycobacterium tuberculosis
We modified it accordingly.
Figure 1. disseminated TB, pericardium- instead of disseminated and pericard
We have revised our figure.
Round 2
Reviewer 2 Report
I have no further comments
I have no comments